# Identification of differentially expressed genes and proteins related to diapause in *Lymantria dispar*: Insights for the mechanism of diapause from transcriptome and proteome analyses

Qing Xie[1,2], Xiaofan Ma[1,2], Yafei Li[1,2], Wenzhuai Ji[1,2], Fengrui Dou[3], Xiue Zhu[4], Juan Shi[1,2]*, Yixia Cao[5]*

1 Beijing Key Laboratory for Forest Pest Control, College of Forestry, Beijing Forestry University, Beijing, China, 2 Sino-France Joint Laboratory for Invasive Forest Pests in Eurasia, College of Forestry, Beijing Forestry University, Beijing, China, 3 Comprehensive Support Center of Hohhot Forestry and Grassland Bureau, Inner Mongolia, China, 4 Guizhou Academy Forestry, Guizhou Province, China, 5 China Certification & Inspection (Group), Inspection Co., Ltd (CCIC), Beijing, China

* shi_juan@263.net (JS); caoyixia@ccic.com (YC)

## Abstract

Spongy moth (*Lymantria dispar* Linnaeus) is a globally recognized quarantine leaf-eating pest. Spongy moths typically enter diapause after completing embryonic development and overwinter in the egg stage. They spend three-quarters of their life cycle (approximately nine months) in the egg stage, which requires a period of low-temperature stimulation to break diapause and continue growth and development. In this study, we explored the molecular mechanism underlying the diapause process in spongy moth. We performed bioinformatics analysis on four Asian populations of spongy moth and one Asian–European hybrid population through a transcriptome analysis combined with proteomics. The results revealed that 1,842 genes were differentially expressed upon diapause initiation, while 264 genes were identified upon diapause termination. Eight diapause-related genes were screened out from the three-level pathways that were significantly enriched by differentially expressed genes at the time of diapause and diapause termination, and the phylogenetic tree and protein three-dimensional structure model were constructed. This study elucidates the diapause mechanism of spongy moth at the gene and protein levels, providing theoretical insights into the early and precise prevention and control of spongy moth. This study can facilitate the development of an efficient, environmentally friendly control system for managing spongy moth populations in the field.

## 1. Introduction

The spongy moth (*Lymantria dispar* Linnaeus, 1758), formerly known as the gypsy moth, is a notorious forest pest native to the Eurasian continent. Currently, the

**Data availability statement:** All relevant data are within the manuscript and its Supporting Information files.

**Funding:** This work was supported by the National Natural Science Foundation of China (Grant No. 32171794). The funders had no role in study design, data collection and analysis, decision to publish, or preparation of the manuscript.

**Competing interests:** The authors have declared that no competing interests exist.

spongy moth has spread to over 50 countries and regions across Europe, the Americas, Africa, and Asia. Due to its highly polyphagous feeding behavior, strong dispersal capability, and potential to cause extensive outbreaks, *L. dispar* poses a significant threat to forest ecosystems in the Holarctic region [1]. This pest has caused numerous outbreaks [2], and its larvae can feed on 300–500 species of broadleaf trees and several species of conifers [3,4], causing substantial ecological and socioeconomic losses. Thus, this forest pest requires focused control measures.

Based on differences in female flight ability and geographical distribution, *L. dispar* can be categorized into European spongy moths (ESM, *L. dispar dispar*) and Asian spongy moths (ASM; a multi-species group represented here by *L. d. asiatica* and *L. d. japonica*). The ASM are collectively designated as the flighted spongy moth complex (FSMC) [5]. Unlike the flightless female ESM [6,7], female FSMC can sustain upward flight [8–10] and exhibit a shorter diapause duration at low temperatures [11], which further enhances their potential for rapid spread and invasion.

To survive adverse environmental conditions such as low temperatures, insects often arrest their development at a specific stage through a process known as diapause [12]. This adaptive strategy ensures population survival and continuity. Diapause can be categorized into obligate and facultative types. In obligate diapause, the process occurs at fixed generations and life stages, while facultative diapause is regulated by external factors like photoperiod, temperature, and nutrition. The spongy moth undergoes obligate diapause after embryonic development, remaining in the egg stage for up to nine months, which accounts for approximately three-quarters of its annual life cycle.

Factors affecting insect diapause include external environmental conditions and internal hormonal changes. Temperature and photoperiod are two major environmental factors affecting insect diapause. Gray et al. measured the respiratory rates of ESM eggs under varying temperatures using the Li-6200 photosynthesis system, demonstrating significant temperature-dependent variations in embryonic metabolic activity [13]. Their work classified spongy moth diapause into three distinct phases: pre-diapause, diapause, and post-diapause, which later researchers collectively termed the the diapause program [14]. Subsequent investigations confirmed that eggs subjected to either insufficient or excessive cold exposure failed to hatch [15], indicating a critical chilling requirement for diapause termination. Subsequent studies established that ESM requires a minimum 75-day chilling exposure at 5°C for diapause termination [16], whereas FSMC resumes development after only 60 days under equivalent conditions [17], indicating lower diapause intensity. This observation has been confirmed in subsequent studies [11,18,19]. Later research by Li (2014) revealed photoperiodic conditions exerted no significant regulatory effect on diapause termination [14].

Internal hormonal signals also play a crucial role in regulating diapause. For example, reduced levels of ecdysteroids such as ecdysone are associated with diapause termination [20], although exogenous ecdysone application appears ineffective in inducing this process [21]. In addition, cryoprotectants like trehalose, glucose, glycerol, mannitol, sorbitol, and inositol have been implicated in maintaining diapause and enhancing cold resistance in FSMC eggs [22].

Current transcriptomic and proteomic studies on diapause processes have been conducted in organisms such as *Bombyx mori* [23–25], *Helicoverpa armigera* [26–28], *Leptinotarsa decemlineata* [29], and *Chilo suppressalis* [30]. However, studies on diapause in the spongy moths have mainly investigated the effects of external environmental factors and metabolic substances, without a systematic exploration at the molecular level.

To fill this research gap, the present study investigated four FSMC populations and one Eurasian hybrid population. We employed transcriptomic techniques (mRNA sequencing) and proteomic methods (TMT labeling) to perform integrative bioinformatics analyses. This study aimed to identify diapause-associated genes and characterize differential gene and protein expression patterns throughout the diapause process in the spongy moth. These findings are expected to provide new insights into the molecular mechanisms of diapause and contribute to the development of molecular-based pest control strategies.

## 2. Materials and methods

### 2.1 Insect strains

The egg masses from the Inner Mongolia (NMG), Shanxi (SX), Liaoning (LN), and Yunnan (YN) populations of *L. dispar* asiatica (FSMC) were collected in the wild from host trees, raised in the Plant Quarantine Laboratory of Beijing Forestry University. In the laboratory, they were reared on an artificial diet until pupation. Upon eclosion, individuals from the same geographical population were paired for mating. In addition, a Eurasian hybrid strain, ONMG, was created by mating male moths from the New Jersey (NJ) population of the United States with female moths from the NMG population of China in the laboratory. The egg masses resulting from these matings were used in the subsequent experiment. Table 1 provides specific information.

During the pre-diapause, diapause, and post-diapause periods, egg masses laid at the same time were selected for the experiment. The division of diapause stages was based on the findings reported by Wang [31]. Specifically, the pre-diapause stage was defined as 0–30 days post-oviposition at 25°C, the diapause stage as 30–90 days post-oviposition at 5°C, and the post-diapause stage as 90 days post-oviposition until larval hatching (approximately 10–30 days) at 25°C. The egg masses from each population during these three periods were respectively named using the abbreviation of the population followed by 1, 2, or 3 (e.g., pre-diapause egg masses of SX were labeled as "SX1"). Each experimental group comprised 500 eggs from each test population, and three replicates were used in the transcriptomic and proteomic analyses. The sampled egg masses were stored at −80°C in a freezer for subsequent experiments.

### 2.2 Transcriptome sequencing and differential expression analysis of spongy moths at different diapause stages

**2.2.1 RNA extraction, quality control, library construction, and sequencing.** The sampled egg masses were thoroughly homogenized in lysis buffer, and the supernatant was collected. RNA was extracted using the TRIzol reagent kit (Invitrogen, Carlsbad, CA, USA), and the purity and concentration of RNA were determined using a Nanodrop 2000 UV spectrophotometer. The quality and integrity of RNA were evaluated using agarose gel electrophoresis and the Agilent 2100 Bioanalyzer (Agilent Technologies, Palo Alto, CA, USA). mRNA was enriched

**Table 1. Location information of the sampling sites for the spongy moth, *Lymantria dispar*.**

| Population Code | Sampling Location | Latitude | Longitude |
|---|---|---|---|
| NMG | Chifeng, Inner Mongolia, China | 41°38′N | 118°53E |
| SX | Shuozhou, Shanxi, China | 39°23′N | 112°66′E |
| LN | Dalian, Liaoning, China | 38°56′N | 121°34′E |
| YN | Fugong, Yunnan, China | 27°09′N | 98°52′E |
| NJ | New Jersey, USA (New Jersey Standard Strain) | 39°41′N | 75°45′W |

using oligo(dT) magnetic beads, followed by fragmentation with fragmentation buffer to obtain fragments approximately 300 bp in size. Single-stranded cDNA was synthesized from mRNA templates by using random hexamers in the presence of reverse transcriptase, followed by second-strand synthesis to form stable double-stranded structures. The sticky ends of double-stranded cDNA were repaired to obtain blunt ends by using the end repair mix, and an "A" base was added to the 3' end for adapter ligation. After adapter ligation, the products were purified and selected based on their size. The size-selected products were subjected to PCR amplification and purification, and subsequently sequenced on the Illumina NovaSeq 6000 sequencing platform (Shanghai Majorbio Bio-pharm Technology Co., Ltd., Shanghai, China).

**2.2.2 Transcriptome de novo assembly and functional annotation.** To ensure the quality of subsequent bioinformatics analyses, we systematically assessed the base distribution and quality fluctuations of reads from all sequencing runs as indicators of sequencing and library construction quality. This included statistical analyses of base composition distribution, base error rate distribution, and base quality distribution for each sequencing cycle. Subsequently, raw sequencing data were filtered to obtain clean data, facilitating downstream analysis for de novo transcriptome studies in the absence of a reference genome. High-quality RNA-seq sequencing data acquisition was followed by de novo assembly to generate singleton sequences and contigs. This process yielded all transcripts from the current transcriptome sequencing experiment.

The obtained transcripts were subjected to the following analyses:

1. Functional annotation: Transcripts were compared against six major databases (i.e., Nonredundant [NR], Swiss-Prot, Pfam, Clusters of Orthologous Groups [COG], Gene Ontology [GO], and Kyoto Encyclopedia of Genes and Genomes [KEGG]) to obtain annotation information for the transcripts and genes;

2. Expression level analysis: RNA-Seq by Expectation-Maximization (RSEM) was performed to quantitatively analyze the expression levels of genes and transcripts;

3. Differential expression analysis: DESeq2 was used to analyze and identify differential expression of genes based on read counts between samples;

4. Enrichment analysis of differential genes in GO and KEGG pathways: GO enrichment analysis was performed using Goatools, and Fisher's exact test was used for precise examination. GO categories with corrected p values (pfdr) ≤ 0.05 were considered significantly enriched. KEGG pathway enrichment analysis was conducted using KEGG Orthology Based Annotation System (KOBAS), and Fisher's exact test was performed for precise examination. The Benjamini–Hochberg (BH) method was used for multiple testing correction to control the false discovery rate. KEGG pathways with corrected p values ≤ 0.05 were considered significantly enriched among differentially expressed genes, from which genes associated with diapause were selected.

**2.2.3 Homology and phylogenetic analysis of diapause-associated genes in spongy moths.** Homology of diapause-associated genes in spongy moths was examined by aligning sequences against the NCBI database (https://blast.ncbi.nlm.nih.gov/Blast), and amino acid sequences with a similarity of >60% were downloaded for phylogenetic analysis. The neighbor-joining (NJ) method was used in MEGA 7.0 to construct phylogenetic trees, with confidence assessed through 1000 bootstrap replicates.

**2.2.4 Construction of three-dimensional models of diapause-associated genes.** The amino acid sequences of diapause-associated genes in spongy moths were aligned on the SwissModel website (https://swissmodel.expasy.org/) to generate protein structure predictions. Models with the highest global model quality estimation values (all exceeding 0.90) were selected as reference templates. Subsequently, protein secondary structures were predicted using the Jpred4 website (https://www.compbio.dundee.ac.uk/jpred4/index_up.html) to enhance the accuracy of protein structure predictions.

At the MolProbity website (http://molprobity.biochem.duke.edu/) and SwissModel website, all three-dimensional models of diapause-associated proteins were evaluated. The Ramachandran plot was used to illustrate whether the dihedral angles of amino acid residues in the main chain of the protein were within reasonable ranges. The MolProbity index was used to evaluate the quality of all atoms within a conformation, including side-chain atoms. MolProbity scoring consider atomic contacts, clashes, bond lengths, angles, and torsion angles.

## 2.3 Proteomic analysis of spongy moths using tandem mass tag labeling

**2.3.1 Protein extraction.** Samples stored at −80°C were ground into powder in liquid nitrogen. Subsequently, each 100 mg of the sample was dissolved in 1 mL of lysis buffer. After sonication for 5 min to remove nucleic acids, the mixture was centrifuged at 3000 rpm for 20 min at 4°C, and the precipitate was collected. The precipitate was then re-suspended in 800 μL of cold acetone to remove impurities. After centrifugation at 3000 rpm for 20 min at 4°C, the precipitate was collected and air-dried. Finally, 600 μL of 8 M urea was added to dissolve the proteins, and the total protein sample was obtained. The protein sample concentration was determined using the Bradford method [32].

**2.3.2 SDS-PAGE electrophoresis.** We extracted 30 μg of the protein sample and diluted it to a final volume of 15 μL. Then, 5 μL of buffer was added, and the mixture was incubated at 95°C water bath for 5–10 min, followed by cooling to room temperature in an ice-water bath and brief centrifugation. The electrophoresis chamber, equipped with a glass plate assembly, was filled with electrode buffer. Approximately 1/4–1/3 of the chamber was filled with the same buffer. Then, 20 μL of protein solution was loaded into the sample wells below the buffer. Sequential electrophoresis runs were performed, including a 20-min run at 80 V for the stacking gel, followed by a 60-min run at 150 V for the resolving gel.

**2.3.3 Reduction, alkylation, and enzymatic digestion.** Triethylammonium bicarbonate buffer (TEAB) was added to the protein sample to achieve a final concentration of 100 mM per 100 μg of protein. Then, tri(2-carboxyethyl) phosphine was added to reach a final concentration of 10 mM, and the mixture was incubated for 60 min. Subsequently, iodoacetamide (IAM) was added to a final concentration of 40 mM, and the reaction was conducted for 40 min in dark conditions. Acetone was then added, and the mixture was precipitated at −20°C for 4 h. After centrifugation at 14,000 rpm for 20 min, the precipitate was collected, and 100 μL of 100 mM TEAB was added to the precipitate. The mixture was then allowed to dissolve completely. Trypsin was added to the solution at an enzyme-to-protein ratio of 1:50, and the mixture was digested at 37°C for at least 12 h.

**2.3.4 Tandem mass tag labeling.** Tandem mass tag (TMT) label reagent (catalog number 90111, ThermoFisher Scientific, Waltham, MA, USA) was mixed with acetonitrile, and the mixture was vortexed and then centrifuged. Then, 100 μg of the aforementioned TMT reagent was added to each sample tube and incubated at room temperature for 2 h. Hydroxylamine was added, and the mixture was incubated for 30 min at room temperature. Finally, the labeled products of each group were obtained and vacuum freeze-dried for later use.

**2.3.5 Ultra-performance liquid chromatography gradient fractionation.** We used the high-pH reversed-phase ultra-performance liquid chromatography (UPLC) fractionation system equipped with the ACQUITY UPLC BEH C18 column to fractionate peptide segments. Phase A consists of 2% acetonitrile (pH 10), whereas phase B consists of 80% acetonitrile (pH 10), with a UV detection wavelength set at 214 nm. The UPLC gradient was programmed as follows: flow rate of 200 μL/min, the gradient started at 0%–5% B for elution from 0 to 17 min, followed by 5%–10% B from 17 to 18 min, 10%–30% B from 18 to 35.5 min, 30%–36% B from 35.5 to 38 min, 36%–42% B from 38 to 39 min, 42%–100% B from 39 to 40 min, and maintained at 100% B until 48 min. The collected 28 fractions were pooled into 14 fractions and then dissolved in a mixture of 2% acetonitrile and 0.1% formic acid buffer solution after negative pressure centrifugation for secondary analysis.

**2.3.6 Liquid chromatography–mass spectrometry analysis.** Using the EASY-nLC liquid chromatography system, peptide segments were separated using mobile phase A (0.1% formic acid) and mobile phase B (100% acetonitrile with 0.1% formic acid).

The liquid chromatography elution gradient was as follows: at a flow rate of 300 nL/min, the gradient started at 5%–38% B for elution from 0 to 30 min, followed by 38%–90% B from 38 to 90 min, and maintained at 90% B until 44 min.

The separated peptide segments were analyzed using the Orbitrap Exploris 480 mass spectrometry system (Thermo, USA) in the data-dependent acquisition mode, with an MS scan range set at 350–1500 m/z. Data acquisition and analysis were performed using Thermo Xcalibur 4.0.

**2.3.7 Database search.** The raw data were analyzed for protein peptide matching and database search identification by using Proteome Discoverer TMSoftware 2.4 software. The database used was all_predicted.pep_unique.fasta. The search parameters were set as follows: trypsin digestion, with up to two missed cleavage sites allowed, a primary mass tolerance of 20 ppm, and a secondary mass tolerance of 0.02 Da.

## 3. Results

### 3.1 Transcriptome sequencing and identification of proteins in spongy moths

To construct cDNA sequencing libraries, we used total RNA extracted from the egg samples collected at the pre-diapause, diapause, and post-diapause stages of spongy moths from five geographic populations. These libraries were sequenced on the Illumina HiSeq platform, and detailed sequencing statistics for each sample are presented in S1 Table (S1 File). The de novo assembly of all clean data from the samples was performed using the software Trinity. The assembled unigenes and transcripts were then merged with all the clean data, leading to mapping rates of 77.99% and 85.64%, respectively. A total of 72,812 unigenes and 100,564 transcripts were obtained, with an average N50 length of 1799 bp. Unigenes accounted for 72.40% of the total transcripts. The length distribution of unigenes (Fig 1A) ranged from 201 to 31,697 bp, with an average sequence length of 933 bp. The majority of the unigenes had lengths ranging from 200 to 500 bp, accounting for 55% of the total count.

For protein identification, 191,240 total spectra were processed, resulting in 26,397 identified spectra. From these, 20,235 peptides were identified, corresponding to 7,644 proteins and 4,499 protein groups. The distribution of peptide lengths is presented in Fig 1B.

### 3.2 Functional annotation

We performed transcriptomic analysis without a reference genome. The assembled unigenes were annotated using six major databases: NR, Swiss-Prot, Pfam, eggNOG, GO, and KEGG (Table 2). Among these unigenes, 5,708 were successfully annotated across all databases, whereas 26,435 were annotated in at least one database. See data S2–S8 Files in the Supplementary Material for detailed information.

All proteins were analyzed using mass spectrometry, and their sequences were compared with four major databases (Pfam, eggNOG, GO, and KEGG) and relevant subcellular localization databases (Table 1). All 4,499 proteins were

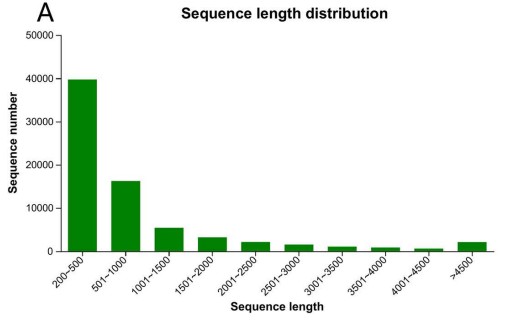
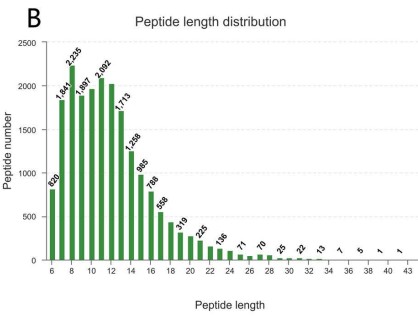

**Fig 1. Distribution of sequence length (A) and peptide length (B).**

**Table 2. Transcriptome and protein annotation statistics of *Lymantria dispar*.**

| Database | Number of Unigenes | Proportion (%) | Protein Number | Proportion (%) |
|---|---|---|---|---|
| NR | 26003 | 35.99 | – | – |
| Swiss-prot | 12448 | 17.23 | – | – |
| Pfam | 13657 | 18.90 | 3994 | 88.78 |
| eggNOG | 20273 | 28.06 | 4372 | 97.18 |
| GO | 14554 | 20.14 | 2658 | 59.08 |
| KEGG | 10288 | 14.24 | 3131 | 69.59 |
| SubCell-Location | – | – | 4499 | 100 |
| Total_anno | 26435 | 36.59 | 4499 | 100 |
| Total | 72255 | 100 | 4499 | 100 |

annotated in at least one database, and 1,426 of them were successfully annotated in all databases. See data S9–S13 Files in the Supplementary Material for detailed information.

On the basis of the homologous alignment of spongy moth unigenes in the NR database, we constructed a distribution chart of NR annotations for species (Fig 2). The results indicated that the most frequently aligned species was *Arctia plantaginis* (23.57%), followed by *H. armigera* (4.30%), *Trichoplusia ni* (3.97%), *Eumeta japonica* (3.48%), and *Spodoptera frugiperda* (3.07%).

A total of 14,554 unigenes and 2,658 proteins from the spongy moths were successfully annotated in the GO database and categorized into three main classes: cellular component (CC), biological process (BP), and molecular function (MF; Fig 3). In the CC category, the cell part had the highest number of unigenes, whereas the cellular anatomical entity had the highest number of proteins. In the BP category, the cellular process entity had the highest count for both unigenes and proteins. In the MF category, catalytic activity had the highest number of unigenes, whereas binding had the highest number of proteins.

In the KEGG database, 10,288 unigenes and 3,131 proteins of the spongy moths were successfully annotated into six metabolic pathways: metabolism, genetic information processing (GIP), environmental information processing (EIP), cellular processes (CP), organismal systems (OS), and human diseases (Fig 4). Among these pathways, except for human diseases, signal transduction had the highest number of unigenes and proteins, followed by transport and catabolism.

## 3.3 Differential gene and protein expression analyses

On the basis of differences in expression levels across various diapause stages, sets of proteins and genes were identified for intergroup differential expression analysis (data S14 and S15 Files). By comparing differentially expressed genes and proteins among the three stages (pre-diapause, diapause, and post-diapause), we screened differentially expressed

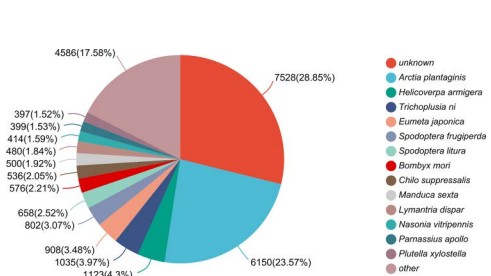

**Fig 2. Species distribution of *Lymantria dispar* in the NR database.**

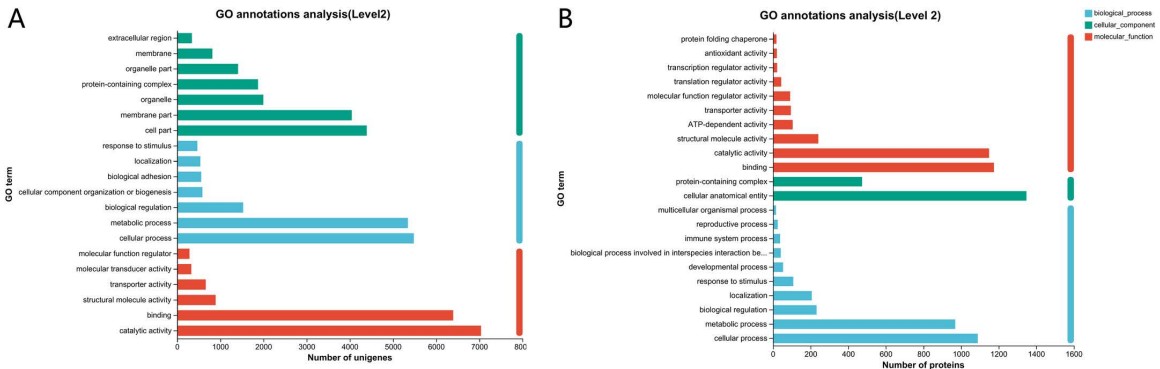

**Fig 3. Transcriptome (A) and proteome (B) GO function distribution statistics of *Lymantria dispar*.**

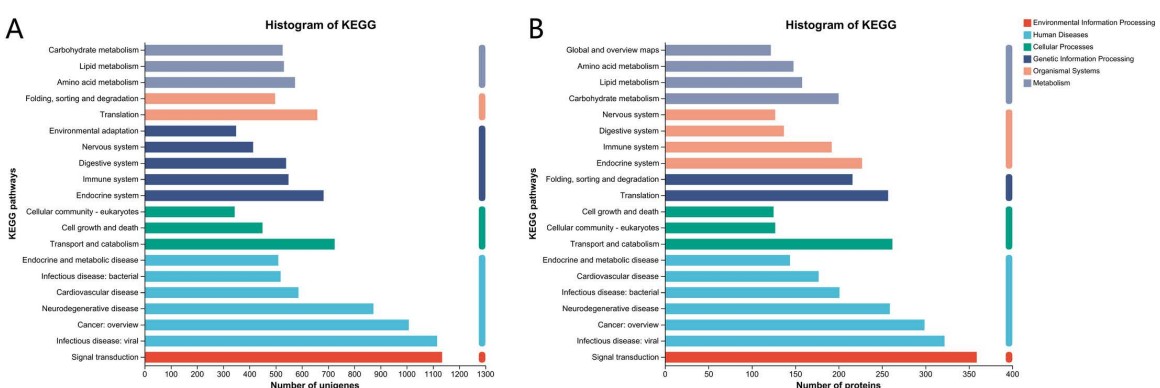

**Fig 4. Transcriptome (A) and proteome (B) KEGG pathway distribution statistics of *Lymantria dispar*.**

genes and proteins involved in the diapause initiation process (transition from pre-diapause to diapause) and the diapause termination process (transition from diapause to post-diapause).

During the diapause initiation process, a total of 1,842 consensus differentially expressed genes (DEGs) were identified across the five populations, including 470 commonly upregulated genes and 465 commonly downregulated genes in diapause compared to pre-diapause, while the remaining genes exhibited differential regulation across populations. Additionally, 1,325 consensus differentially expressed proteins (DEPs) were identified, comprising 822 commonly upregulated proteins and 264 commonly downregulated proteins.

During the diapause termination process, 264 consensus DEGs were identified across the five spongy moth populations, including 87 commonly upregulated genes and 58 commonly downregulated genes in post-diapause compared to diapause. Furthermore, 442 consensus DEPs were identified, consisting of 123 commonly upregulated proteins and 83 commonly downregulated proteins. Fig 5 presents the numbers of differentially expressed genes and proteins at different diapause stages for each population. Detailed information can be found in Supplementary Material S16–S19 Files.

### 3.4 Population-specific diapause-related genes and pathway enrichment analysis

KEGG enrichment analysis of population-specific DEGs during diapause initiation process and diapause termination process revealed distinct regulatory strategies among geographical populations (S1–S5 Figs in S1 File). Northern populations (LN, NMG) preferentially activated stress response pathways during diapause initiation process: the LN

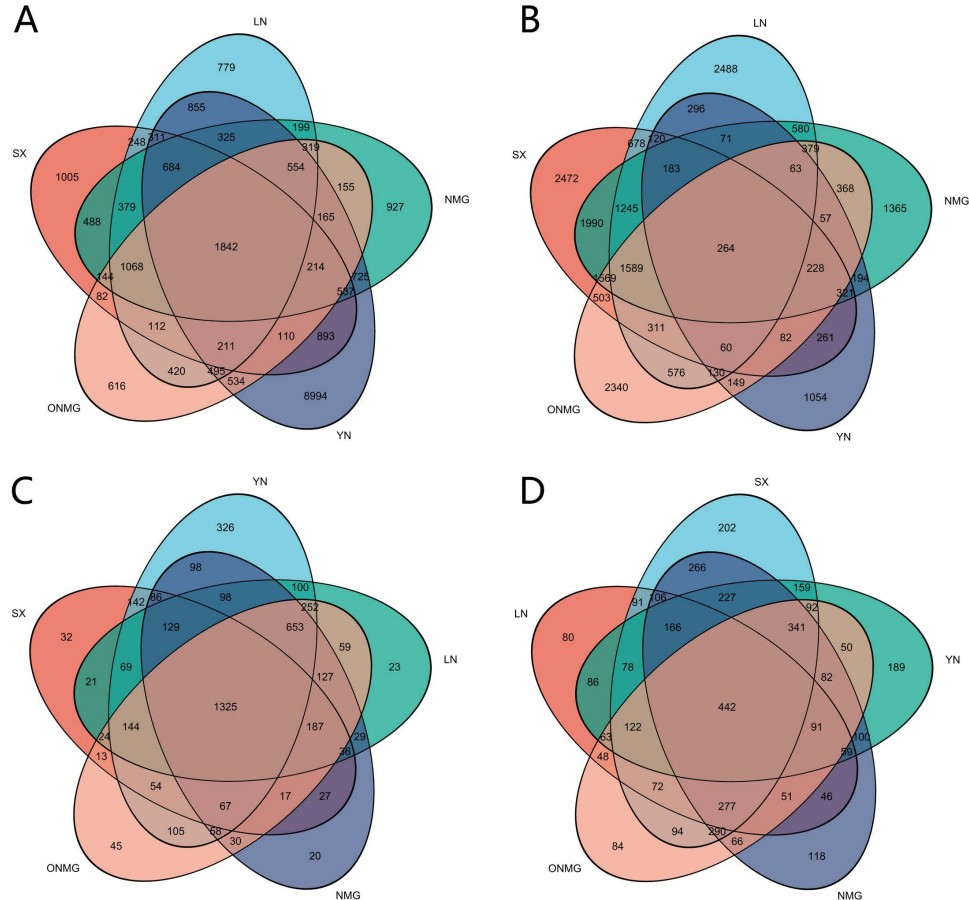

**Fig 5. Differentially expressed genes and proteins in different diapause stages of *Lymantria dispar*.** (A) Differentially expressed genes when five *L. dispar* populations enter diapause; (B) Differentially expressed genes during diapause termination in five *L. dispar* populations; (C) Differentially expressed proteins when five *L. dispar* populations enter diapause; (D) Differentially expressed proteins during diapause termination in five *L. dispar* populations.

population regulated neuroendocrine plasticity through the Neurotrophin signaling pathway, while the NMG population enhanced immune tolerance via the Complement and coagulation cascades pathway, simultaneously suppressing the Cell cycle pathway to achieve developmental arrest. During diapause termination process, the LN population shifted to the Renin-angiotensin system to regulate osmotic balance, whereas the NMG population maintained immune homeostasis through sustained activity of the Complement and coagulation cascades pathway, indicating their adaptation to cold-arid environments through neuro-immune coordination. The transitional population (SX) adapted to energy demands during diapause initiation process by regulating Galactose metabolism pathway, while activating the Vascular smooth muscle contraction pathway during termination to promote fluid circulation recovery, reflecting metabolic plasticity-driven environmental adaptation. The southern population (YN) strengthened Mismatch repair during initiation process to maintain genomic stability and activated Olfactory transduction during termination process to precisely regulate hatching behavior, demonstrating refined adaptation to humid-tropical environments. The Eurasian hybrid population (ONMG) integrated Oxidative phosphorylation and N-Glycan biosynthesis during diapause initiation process, and synergistically upregulated Glycolysis/Gluconeogenesis and the Citrate cycle during termination process, forming a metabolic network cascade that provided energy metabolism advantages for ecological niche expansion.

This hierarchical pathway divergence not only elucidates the molecular biogeography of diapause regulation but also reveals evolutionary trade-offs in key biological functional modules: northern populations optimized survival costs through signal transduction, transitional and southern populations ensured genetic adaptation via genomic stability, and hybrid populations enhanced niche adaptability through metabolic network expansion.

### 3.5 Selection of diapause-associated genes in spongy moths

The differentially expressed genes were analyzed using the KEGG pathway analysis. During the diapause initiation stage, the differentially expressed genes shared by the five populations were annotated to 214 KEGG metabolic pathways (Fig 6A), excluding human diseases. Among these, 21 were associated with CP, 28 with EIP, 18 with GIP, 67 with metabolism, and 80 with OS. During the diapause termination stage, the differentially expressed genes shared by the five populations were annotated to 58 KEGG metabolic pathways (Fig 6B), excluding human diseases. Among these, 11 were associated with CP, 10 with EIP, 5 with GIP, 16 with metabolism, and 16 with OS.

A total of 52 pathways were significantly enriched by DEGs during both the diapause initiation process and termination stages process (P<0.05), with detailed information provided in Supplementary Material S20 and S21 Files. Based on current research evidence [33–39], three pathways with established regulatory roles in diapause were selected for subsequent investigation: glutathione metabolism, citrate cycle (TCA cycle), and alanine, aspartate and glutamate metabolism pathways.

10 differentially expressed genes were significantly enriched in the glutathione metabolism pathway (S6 and S7 Figs in S1 File), where glutathione acts as a crucial intracellular antioxidant. In this pathway, during the diapause initiation stage, the expression of *GST* was significantly upregulated, enhancing the synthesis of glutathione S-transferase and reducing total glutathione content. However, during the diapause termination stage, the expression of *GCLC* was significantly upregulated, promoting the synthesis of glutamate–cysteine ligase and increasing the total glutathione content.

Furthermore, 3 differentially expressed genes were significantly enriched in the citrate cycle pathway (S8 Fig in S1 File), where the expression levels of *IDH1*, *IDH2*, and *icd* were markedly downregulated during the diapause initiation stage in spongy moth eggs, suppressing the activity of citrate cycle and modulating the rate of energy release.

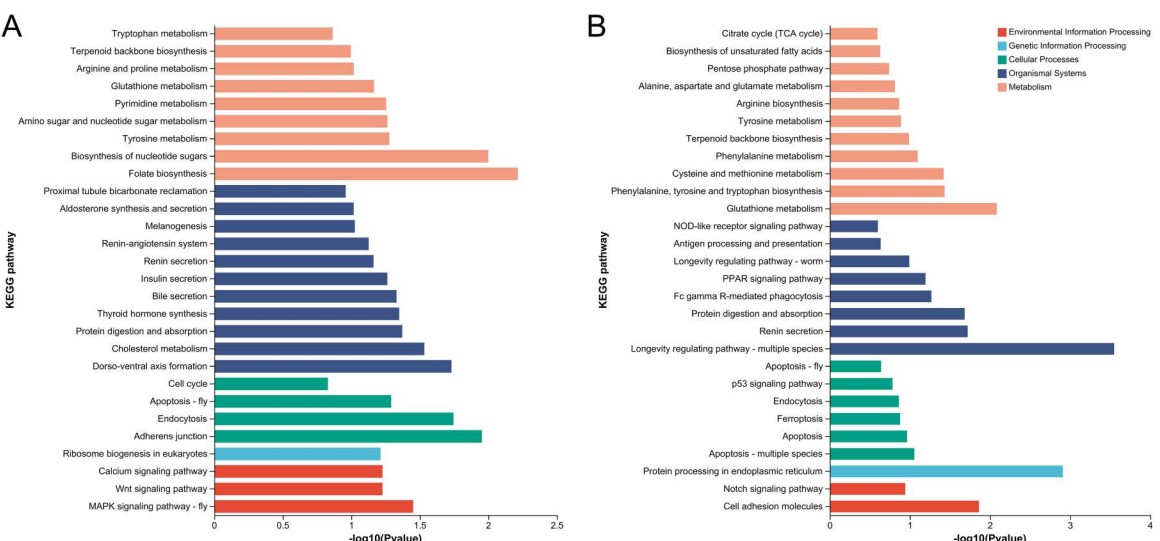

**Fig 6. KEGG Enrichment Analysis of Shared Differentially Expressed Genes Across Diapause Stages in Five *Lymantria dispar* Populations.** (A) Shared differentially expressed genes in the Diapause Initiation Process; (B) Shared differentially expressed genes in the Diapause Termination Process.

Moreover, 4 differentially expressed genes were significantly enriched in the alanine, aspartate, and glutamate metabolism pathway (S9 and S10 Figs in S1 File), where spongy moth eggs exhibited significantly increased expressions of *GLUD1_2* and *gdhA* during the diapause initiation stage, facilitating the synthesis of mitochondrial glutamate dehydrogenase and formation of aspartate entering the urea cycle. However, during the diapause termination stage, the expression of *GOT1* was significantly upregulated, enhancing the synthesis of aspartate transaminase and leading to the formation of glutamate and oxaloacetate, thus reducing urea levels.

On the basis of the enrichment of these differentially expressed genes during the initiation and termination stages of diapause, we speculate that these genes may be involved in the diapause process of spongy moths. Thus, we next analyzed eight diapause-associated genes, namely *GST*, *GCLC*, *IDH1*, *IDH2*, *icd*, *GLUD1_2*, *gdhA*, and *GOT1*.

### 3.6 Diapause-associated gene homology and phylogenetic analysis

This study investigated the evolutionary relationships of eight diapause-related genes in *L. dispar* through phylogenetic analysis (Fig 7).

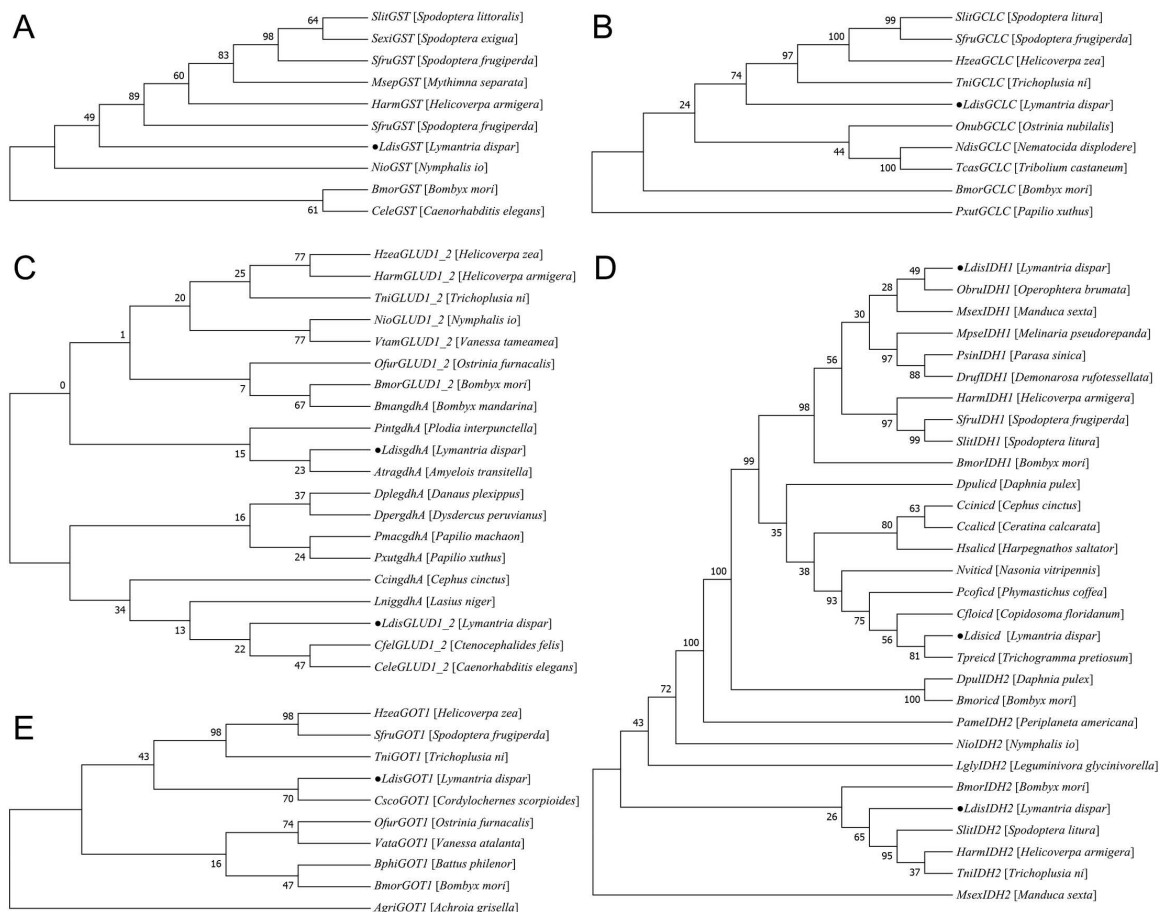

**Fig 7. Phylogenetic tree of the amino acid sequences of the eight diapause-associated genes in *Lymantria dispar*.** Black dots indicate the diapause-associated genes in *L. dispar*. (A) Phylogenetic reconstruction of *LdGST*; (B) Evolutionary analysis of *LdGCLC*; (C) Phylogenetic relationships of *LdGLUD1_2* and *LdgdhA*; (D) Cladogram of *LdIDH1*, *LdIDH2*, and *Ldicd*; (E) Phylogenetic topology of *LdGOT1*.

(1) The phylogenetic analysis of *LdGST* (Fig 7A) revealed that this gene formed a distinct clade with *GST* genes from Noctuidae species, supporting its conserved evolutionary pattern within Lepidoptera.

(2) *LdGCLC* (Fig 7B) clustered with the GCLC gene of *T. ni* and further grouped with *H. zea* GCLC to form an evolutionary branch, indicating high conservation among Lepidoptera. Significant evolutionary divergence was observed between *LdGCLC* and homologs from *Nematocida displodere* and *Tribolium castaneum*.

(3) Phylogenetic analysis of *LdGLUD1_2* and *LdgdhA* (Fig 7C) demonstrated that *LdGLUD1_2* formed an independent clade with *GLUD1_2* genes from *Ctenocephalides felis* and *Caenorhabditis elegans*. *LdgdhA* exhibited the closest evolutionary relationship with the *gdhA* gene of *Amyelois transitella*, suggesting functional conservation in Lepidoptera.

(4) The phylogenetic reconstruction of *LdIDH1*, *LdIDH2*, and *Ldicd* (Fig 7D) showed that both *LdIDH1* and *LdIDH2* clustered within the Lepidoptera branch, forming a group with *IDH* genes from *B. mori*, *H. armigera*, and *S. litura*. In contrast, *Ldicd* grouped with the *icd* gene of *Trichogramma pretiosum*.

(5) Analysis of *LdGOT1* (Fig 7E) revealed that this gene formed an independent evolutionary branch with *Cordylochernes scorpioides* GOT1, which subsequently clustered as a sister group to an evolutionary cluster containing *H. zea*, *S. frugiperda*, and *T. ni*.

### 3.7  Prediction of three-dimensional structures of diapause-associated proteins in spongy moths

The constructed structural models of diapause-associated proteins revealed that GST comprises eight α-helices and four β-strands; GCLC consists of 25 α-helices and 18 β-strands; IDH1 features 13 α-helices and 10 β-strands; IDH2 contains 12 α-helices and 10 β-strands; icd includes 11 α-helices and 12 β-strands; GLUD1_2 possesses 18 α-helices and 13 β-strands; gdhA exhibits three α-helices and a single β-strand; and GOT1 presents 13 α-helices and 10 β-strands (Fig 8).

   The stereochemical quality of protein three-dimensional structural models was evaluated using Ramachandran plots, with all models demonstrating >99% of amino acid residues in allowed regions (S11 Fig in S1 File). Model quality was further validated through MolProbity scoring incorporating clashscore, rotamer outliers, and Ramachandran plot statistics (S2 Table in S1 File).

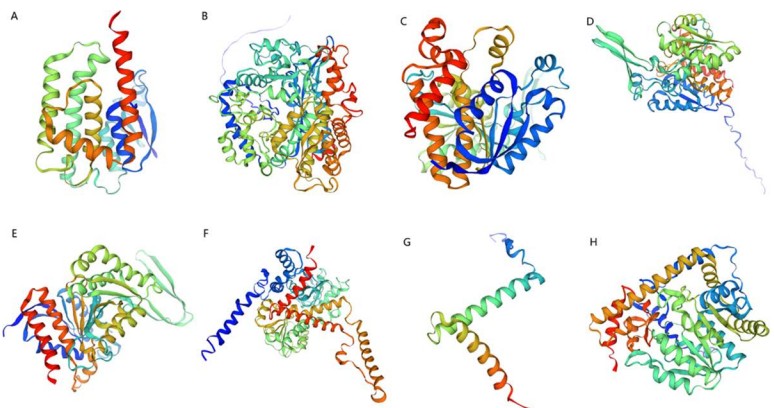

**Fig 8.  Three-dimensional protein structure prediction.** GST (A), GCLC (B), IDH1 (C), IDH2 (D), icd (E), GLUD1_2 (F), gdhA (G), GOT1 (H).

## 4. Discussion

This study integrates transcriptomic and proteomic data to investigate molecular regulatory mechanisms underlying diapause in *Lymantria dispar*. Comparative analysis of DEGs during diapause initiation process (diapause vs. pre-diapause) and termination process (post-diapause vs. diapause) identified 52 significantly enriched KEGG pathways. Three core metabolic pathways – glutathione metabolism, citrate cycle (TCA cycle), and alanine, aspartate and glutamate metabolism pathways – were prioritized for detailed investigation. Eight key regulatory genes (*GST*, *GCLC*, *IDH1*, *IDH2*, *icd*, *GLUD1_2*, *gdhA*, *GOT1*) were identified through pathway analysis. These genes exhibit pathway-specific functional significance in diapause regulation, as detailed below through comparative analysis with other insect systems and population-level variations.

The glutathione metabolism pathway plays a pivotal role in insect diapause regulation [33–35]. Glutathione (GSH), as the primary intracellular antioxidant, is essential for eliminating reactive oxygen species (ROS), detoxifying peroxides, and repairing damaged cellular components [40]. In our study, we observed an upregulation of *GST* during diapause initiation process in *L. dispar* eggs, while *GCLC* expression was significantly elevated during diapause termination process. These findings suggest that *L. dispar* may initially upregulate *GST* to remove excess ROS through detoxification, leading to a transient decline in total GSH levels. During late diapause, GSH biosynthesis is likely enhanced to restore redox balance and resume metabolic activity. Similar redox regulatory patterns have been reported in other insects, including *Ostrinia nubilalis* [41], *Chlosyne lacinia* [34], *Artemia* [42], and *B. mori* [43], highlighting the central role of the glutathione pathway in maintaining cellular homeostasis during diapause.

The citrate cycle (TCA), a core pathway of cellular energy metabolism, is also tightly regulated during diapause [36,37]. In our data, genes including *IDH1*, *IDH2*, and *icd* were significantly downregulated during diapause initiation process, suggesting a suppression of TCA cycle activity and a shift toward a low-energy metabolic state [37,44,45]. This aligns with observations in multiple species where *IDH* activity is reduced under diapause conditions, such as in *Rhagoletis pomonella* [46], *B. mori* [47], and *Coccinella septempunctata* [36]. However, some species (e.g., *C. lacinia* [34], *Exorista civilis* [48]) show increased *IDH* levels during diapause stage, which may reflect the antioxidant role of IDH enzymes [49].

Regulation of the alanine, aspartate, and glutamate metabolism pathway is critical for maintaining metabolic balance during diapause [38,39]. This pathway interconnects amino acid and energy metabolism and contributes to nitrogen balance [39,50,51]. We observed upregulation of *GLUD1_2* and *gdhA* during diapause initiation process, coinciding with a decrease in glutamate and an increase in aspartate levels. This likely activates the urea cycle as a mechanism to manage excess nitrogen due to reduced protein synthesis. This is consistent with findings in *H. armigera*, which accumulates urea to counter cold stress during diapause [38]. During diapause termination process, upregulation of *GOT1* may help restore glutamate levels and repress the urea cycle, facilitating cellular proliferation and growth. Together, these dynamic shifts in amino acid metabolism underscore the central role of this pathway in diapause regulation.

Our study also employed a comparative population genomics strategy, incorporating *L. dispar* populations from Inner Mongolia (NMG), Liaoning (LN), Shanxi (SX), Yunnan (YN), and a Eurasian hybrid strain (ONMG). Population-level differences in gene regulation suggest varied diapause strategies shaped by environmental and genetic factors. For instance, northern populations (e.g., NMG, LN) appear to rely more on endocrine and energy metabolism modulation to survive cold conditions, while southern populations (e.g., YN) emphasize antioxidant and cellular protection mechanisms. The Eurasian hybrid population displayed mixed regulatory features, indicative of underlying genetic plasticity in diapause control. These findings underscore the dual influence of environmental adaptation and genetic background in shaping diapause strategies.

Currently, research on *L. dispar* diapause remains largely focused on external environmental factors (e.g., temperature, photoperiod) and metabolic profiling, with limited exploration at the molecular and genetic level. This study integrates transcriptomic and proteomic data and applies a multi-population approach to systematically identify candidate genes involved in diapause regulation. These findings provide a foundation for future functional studies and offer potential targets for genetic or RNAi-based pest control strategies.

From an applied perspective, our results have clear practical relevance. The identification of diapause-related genes offers a molecular basis for precision pest management. Moreover, the observed inter-population regulatory differences highlight the need for region-specific control strategies. In the context of global climate change and species invasions, understanding how insects adapt at the molecular level is critical for forecasting pest outbreaks and developing adaptive management plans [52,53].

Despite these advances, several challenges remain. First, diapause is a complex process involving multiple interacting pathways [54,55]. This study focused on three key metabolic routes, but other potentially important regulatory networks warrant further investigation. Second, the molecular functions of candidate genes require validation through gene knock-out, protein interaction, and metabolomic approaches. While the Eurasian hybrid population revealed genetic plasticity in diapause control, the limited sample size calls for broader geographic and genomic sampling, particularly including ESM, to fully elucidate interlineage variation.

In conclusion, this study identified eight key diapause-associated genes (*GST*, *GCLC*, *IDH1*, *IDH2*, *icd*, *GLUD1_2*, *gdhA*, *GOT1*) through integrated transcriptomic and proteomic analyses across multiple populations. Structural and functional predictions of these proteins further supported their roles in diapause regulation. Our findings enhance the understanding of the molecular basis of insect diapause and offer a theoretical framework for the development of targeted and efficient pest control strategies. Future research will focus on mapping gene regulatory networks and dissecting their physiological roles to support comprehensive *L. dispar* management.

## Supporting information

**S1 File. Supplementary tables and figures.**
(DOCX)

**S2 File. NR annotation information table-transcriptome functional annotation.**
(XLSX)

**S3 File. NR annotated species distribution table-transcriptome functional annotation.**
(XLSX)

**S4 File. Swiss-prot annotation information table-transcriptome functional annotation.**
(XLSX)

**S5 File. Pfam annotation information table-transcriptome functional annotation.**
(XLSX)

**S6 File. EggNOG classification statistics table-transcriptome functional annotation.**
(XLSX)

**S7 File. GO classification statistics table-transcriptome functional annotation.**
(XLSX)

**S8 File. KEGG classification statistics table-transcriptome functional annotation.**
(XLSX)

**S9 File. Pfam annotation information table-proteome functional annotation.**
(XLSX)

**S10 File. EggNOG classification statistics table-proteome functional annotation.**
(XLSX)

**S11 File. GO classification statistics table-proteome functional annotation.**
(XLSX)

**S12 File. KEGG classification statistics table-proteome functional annotation.**
(XLSX)

**S13 File. Subloc annotation information table-proteome functional annotation.**
(XLSX)

**S14 File. Genome-wide gene expression dataset across five populations and all diapause stages.**
(CSV)

**S15 File. Comprehensive protein expression dataset across five populations and all diapause stages.**
(CSV)

**S16 File. GO enrichment analysis statistical table of DEGs upon diapause initiation.**
(XLSX)

**S17 File. GO enrichment analysis statistical table of DEGs upon diapause termination.**
(XLSX)

**S18 File. KEGG enrichment analysis statistical table of DEGs upon diapause initiation.**
(XLSX)

**S19 File. KEGG enrichment analysis statistical table of DEGs upon diapause termination.**
(XLSX)

**S20 File. KEGG pathway enrichment during diapause initiation process in *Lymantria dispar.***
(XLSX)

**S21 File. KEGG pathway enrichment during diapause termination process in *Lymantria dispar.***
(XLSX)

## Acknowledgments

We would like to thank Yixin Qiu (Beijing Forestry University), Yanyi Lu (Beijing Forestry University) and Yuxuan Li (Beijing Forestry University) for assistance in raising spongy moth.

## Author contributions

**Conceptualization:** Qing Xie, Xiaofan Ma, Yafei Li, Juan Shi.

**Formal analysis:** Qing Xie, Wenzhuai Ji, Fengrui Dou.

**Funding acquisition:** Juan Shi.

**Investigation:** Qing Xie, Xiaofan Ma, Yafei Li.

**Methodology:** Qing Xie, Xiaofan Ma, Yafei Li.

**Project administration:** Juan Shi, Yixia Cao.

**Resources:** Xiue Zhu.

**Supervision:** Juan Shi, Yixia Cao.

**Visualization:** Qing Xie.

**Writing – original draft:** Qing Xie.

**Writing – review & editing:** Qing Xie, Juan Shi.

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
