## [Decision Letter · Decision Letter 0]

25 Feb 2025

PONE-D-24-55985Identification of Differentially Expressed Genes and Proteins Related to Diapause in Lymantria Dispar: Insights for the Mechanism of Diapause from Transcriptome and Proteome AnalysesPLOS ONE

Dear Dr. Shi,

Thank you for submitting your manuscript to PLOS ONE. After careful consideration, we feel that it has merit but does not fully meet PLOS ONE’s publication criteria as it currently stands. Therefore, we invite you to submit a revised version of the manuscript that addresses the points raised during the review process as mainly described here.  Please submit your revised manuscript by Apr 11 2025 11:59PM. If you will need more time than this to complete your revisions, please reply to this message or contact the journal office at plosone@plos.org . Please include the following items when submitting your revised manuscript:

We look forward to receiving your revised manuscript.

Kind regards,

Munir Ahmad, PhD

Academic Editor

PLOS ONE

Journal Requirements:

“National Natural Science Foundation of China, grant number 32171794”

Additional Editor Comments:

Reorganize the content and necessitate substantial revisions to the manuscript, sample collection procedure need elaboration with dataset of all stages of diapause, discussion should focus five selected species compared with availability of dataset for all; may be as supplementary file, should focus the respective genes of interest relevant rather than being general and improvement of discussion as kindly suggested by the reviewer before submission, may be followed.

Reviewers' comments:

Reviewer's Responses to Questions

**Comments to the Author**

1. Is the manuscript technically sound, and do the data support the conclusions?

Reviewer #1: Yes

2. Has the statistical analysis been performed appropriately and rigorously? 

Reviewer #1: Yes

3. Have the authors made all data underlying the findings in their manuscript fully available?

Reviewer #1: Yes

4. Is the manuscript presented in an intelligible fashion and written in standard English?

Reviewer #1: Yes

5. Review Comments to the Author

Reviewer #1: In this study, the authors employed transcriptomic and proteomic analyses to investigate the molecular mechanisms underlying the diapause process in the spongy moth, identifying several differentially expressed genes. This research contributes to a comprehensive understanding of the molecular basis of spongy moth diapause. However, numerous descriptions of this manuscript were perplexing, and certain deficiencies in the article necessitate thorough revision.

1. The writing is imprecise and confusing, with numerous inappropriate statements that hinder the effective communication of the message. So i think the authors should reorganize the content and necessitate substantial revisions to the manuscript.

2. In the Materials and Methods section, line 128, the authors mention that eggs during pre-diapause, diapause, and post-diapause periods were selected as samples. However, i suggest specifying the exact timing (e.g., the specific day of egg development) for greater clarity.

3. The authors conducted an analysis of the transcriptome and proteome across five species of spongy moth. What was the purpose of chose these five species? After all, there is no relevant analysis in the results section and no discussion in the discussion section.

4. According to the author's description, transcriptomic analyses were conducted on five species. However, only one dataset (including Q30, mapping rates, etc.) is presented in the results. Why? It is advisable to include detailed data in the supplementary materials or upload them to the NCBI database for comprehensive accessibility.

5. In the results section, the authors described "The period from pre-diapause to mid-diapause was termed as the diapause initiation stage, whereas the period from mid-diapause to post-diapause was termed as the diapause termination stage." Which period constitutes the "diapause maintenance stage."? Additionally, the Materials and Methods section indicates that diapause egg samples were selected, yet the results lack data pertaining to the diapause stage.

6. In the results, lines 341-351, the authors said "470 showing an upregulation and 465 showing a downregulation... 822 being upregulated and 264 being downregulated...". It is necessary to clarify the reference group used for these upregulated and downregulated comparisons.

7. In line 367, the full name of "CP" should be provided.

8. In the results section titled "3.4 Selection of Diapause-Associated Genes in Spongy Moths", it should added detailed data, such as figures, to substantiate the description.

9. In the results section titled "3.4 Selection of Diapause-Associated Genes in Spongy Moths", why authors focus and selected these genes (GST, GCLC, IDH1, etc.) for in-depth analysis? In deed, there are many differentially expressed genes. Is this selection based on subjective choice or prior analyses? Moreover, i recommended that the authors should integrate transcriptomic and proteomic analyses to explore the diapause related genes.

10. In the section titled "3.5 Diapause-Associated Gene Homology and Phylogenetic Analysis", the evolutionary tree results are improperly described. Why the authors described the evolutionary relationships of the genes in different families? The authors should describe the evolutionary relationship of each gene across species. Furthermore, the description beginning with "The amino acid...various closely related species" has already been detailed in the Materials and Methods section and should be removed for redundancy.

11. In the section of "3.6 Prediction of Three-Dimensional Structures of Diapause-Associated Proteins in Spongy Moths", Figure 8 and Table 3 pertain to the reliability verification of the tertiary structure model. These can be moved to the supplementary materials.

12. The discussion section is muddled. I would recommend authors to summarize the topic based on this research, and revising a large portion of the discussion section.

6. PLOS authors have the option to publish the peer review history of their article (what does this mean? ). If published, this will include your full peer review and any attached files.

**Do you want your identity to be public for this peer review?** For information about this choice, including consent withdrawal, please see our Privacy Policy .

Reviewer #1: No

---

## [Author Response · Author response to Decision Letter 1]

18 Apr 2025

Comment 1:

The writing is imprecise and confusing, with numerous inappropriate statements that hinder the effective communication of the message. So i think the authors should reorganize the content and necessitate substantial revisions to the manuscript

Answer:

We sincerely thank the reviewer for their attention to the writing quality of our manuscript. To ensure the scientific accuracy and clarity of our study, we have made substantial revisions throughout the entire manuscript. These include language refinement, structural reorganization, clarification of key concepts, and optimization of logical coherence. The specific modifications are summarized as follows:

1. Introduction

We made extensive and in-depth revisions to the Introduction section, rewriting the majority of its content to improve clarity and enhance logical flow. Key improvements include:

(1) Structural Reorganization: In the original version, information about the ecological background of Lymantria dispar, its damage, and previous research on diapause was scattered and lacked cohesion. We have now restructured the Introduction to follow a clear logical sequence: background - current knowledge - knowledge gaps - research objectives, effectively conveying the research rationale and importance.

(2) Concept Clarification: Key biological terms such as the classification of L. dispar and the types of diapause have been revised for greater precision. To improve conceptual continuity, the explanation of diapause types now precedes the discussion of its occurrence in L. dispar.

(3) Language Refinement: We have streamlined redundant or repetitive descriptions. For instance, previously separate paragraphs on environmental factors influencing diapause have now been merged into a coherent section. Each paragraph is now centered around a single topic and begins with a topic sentence to strengthen inter-paragraph logic.

(4) Stylistic Improvement: We have refined wording for greater professionalism and fluency. Ambiguous or repetitive terms were replaced with more accurate expressions (e.g., “numerous outbreaks” → “extensive outbreaks”).

2. Materials and Methods

We clarified previously vague content, such as the exact definition of the pre-diapause, diapause, and post-diapause stages. Specifically, we added the number of days and temperature conditions corresponding to each stage (Lines 118–121). Furthermore, we replaced inconsistent descriptions of developmental transitions with precise terms: diapause initiation process (transition from pre-diapause to diapause) and the diapause termination process (transition from diapause to post-diapause), avoiding confusion present in the original version (Lines 329–333).

3. Results

The Results section was substantially revised, with many additions, deletions, and reorganization. Major improvements include:

(1) Population-specific analysis: We added new analyses of differentially expressed genes (DEGs) and pathways across five geographically distinct populations. This highlights the combined influence of environmental and genetic factors in diapause regulation and strengthens the significance of our multi-population design.

(2) Enhanced data transparency: New supplementary data were added, including transcriptome quality metrics for all five populations, developmental stage-specific gene/protein expression profiles, details of 52 significantly enriched third-level KEGG pathways, and KEGG pathway maps during diapause initiation and termination.

(3) Phylogenetic analysis revision: We reconstructed the phylogenetic trees following standard practices and rewrote all related descriptions based on species-specific evolutionary relationships of the candidate genes.

(4) Figure updates: We added new figures to support KEGG enrichment and gene selection rationale, and moved tertiary structure validation figures to the supplementary materials for clarity.

4. Discussion

The Discussion section was completely rewritten to improve logical flow, coherence, and scientific depth. Major revisions include:

(1) Summary of major findings: We now provide concise discussion of the biological significance of the three main metabolic pathways (glutathione metabolism, TCA cycle, alanine, aspartate and glutamate metabolism), integrating our findings with existing literature.

(2) Population-level insights: We expanded the discussion to compare diapause regulatory strategies across different geographic populations, revealing adaptation mechanisms to varying environmental pressures.

(3) Application and future outlook: A new paragraph discusses the applied relevance, current limitations, and directions for future research.

All changes are highlighted in yellow in the revised manuscript. We also provide original and revised text excerpts with corresponding line numbers in our point-by-point response. Once again, we sincerely thank the reviewer for their constructive and insightful suggestions, which have significantly improved the clarity and quality of our manuscript.

Comment 2:

In the Materials and Methods section, line 128, the authors mention that eggs during pre-diapause, diapause, and post-diapause periods were selected as samples. However, i suggest specifying the exact timing (e.g., the specific day of egg development) for greater clarity.

Answer:

Thank you for this thoughtful suggestion. We have now clarified the specific timing of each diapause stage in the revised manuscript. The number of days and corresponding temperature conditions are now clearly defined in the Materials and Methods section (Lines 118–121), as follows:

Specifically, the pre-diapause stage was defined as 0–30 days post-oviposition at 25°C, the diapause stage as 30–90 days post-oviposition at 5°C, and the post-diapause stage as 90 days post-oviposition until larval hatching (approximately 10–30 days) at 25°C.

Comment 3:

The authors conducted an analysis of the transcriptome and proteome across five species of spongy moth. What was the purpose of chose these five species? After all, there is no relevant analysis in the results section and no discussion in the

discussion section.

Answer:

Thank you for raising this important point. In our study, we selected five geographically distinct populations of L. dispar (from Inner Mongolia, Shanxi, Liaoning, Yunnan, and one Asian-European hybrid population) for transcriptomic and proteomic analysis. However, in the original version, we only focused on the shared DEGs and did not elaborate on the population-specific differences in either the Results or Discussion sections.

We have now revised the manuscript to address this gap. Below are our detailed justifications and updates:

1. Rationale for Population Selection

Our choice of populations was based on three main scientific considerations:

(1) To investigate how diapause regulation varies across distinct environmental conditions;

(2) To assess whether genetic background influences the expression of diapause-related genes;

(3) To explore the regulatory plasticity of diapause in a hybrid population.

The four Asian-type populations span a north-to-south geographic gradient within China, representing a natural climate diversity. Meteorological data from the China Meteorological Administration (https://data.cma.cn) show the following average winter temperatures (2014–2024): Inner Mongolia: –16.2°C; Liaoning: –13.7°C; Shanxi: –5.8°C; Yunnan: 7.1°C.

These gradients likely affect the length and intensity of low-temperature exposure required to terminate diapause. By comparing gene expression patterns among these populations, we aim to identify how local adaptation may shape diapause regulation.

In addition, the inclusion of a hybrid population (Inner Mongolia male × New Jersey female) allowed us to explore the potential impact of genetic background on diapause gene regulation. Previous studies have shown that Asian-type moths typically require shorter chilling periods to terminate diapause compared to European-type moths. Although we were unable to include a pure European population due to sample limitations, the hybrid population provides valuable insights into potential maternal or biparental inheritance effects in diapause regulation.

2. New Results Added to the Manuscript

We have added population-specific DEG analyses during diapause initiation and termination. We identified uniquely expressed genes for each population and conducted KEGG enrichment analysis to determine the major pathways involved (Lines 356–388). These additions strengthen the multi-population comparative framework of our study.

3.4 Population-Specific Diapause-Related Genes and Pathway Enrichment Analysis

KEGG enrichment analysis of population-specific DEGs during diapause initiation process and diapause termination process revealed distinct regulatory strategies among geographical populations (File S1, Supplementary Figures 1-5). Northern populations (LN, NMG) preferentially activated stress response pathways during diapause initiation process: the LN population regulated neuroendocrine plasticity through the Neurotrophin signaling pathway, while the NMG population enhanced immune tolerance via the Complement and coagulation cascades pathway, simultaneously suppressing the Cell cycle pathway to achieve developmental arrest. During diapause termination process, the LN population shifted to the Renin-angiotensin system to regulate osmotic balance, whereas the NMG population maintained immune homeostasis through sustained activity of the Complement and coagulation cascades pathway, indicating their adaptation to cold-arid environments through neuro-immune coordination. The transitional population (SX) adapted to energy demands during diapause initiation process by regulating Galactose metabolism pathway, while activating the Vascular smooth muscle contraction pathway during termination to promote fluid circulation recovery, reflecting metabolic plasticity-driven environmental adaptation. The southern population (YN) strengthened Mismatch repair during initiation process to maintain genomic stability and activated Olfactory transduction during termination process to precisely regulate hatching behavior, demonstrating refined adaptation to humid-tropical environments. The Eurasian hybrid population (ONMG) integrated Oxidative phosphorylation and N-Glycan biosynthesis during diapause initiation process, and synergistically upregulated Glycolysis/Gluconeogenesis and the Citrate cycle during termination process, forming a metabolic network cascade that provided energy metabolism advantages for ecological niche expansion.

This hierarchical pathway divergence not only elucidates the molecular biogeography of diapause regulation but also reveals evolutionary trade-offs in key biological functional modules: northern populations optimized survival costs through signal transduction, transitional and southern populations ensured genetic adaptation via genomic stability, and hybrid populations enhanced niche adaptability through metabolic network expansion.

3. Expanded Discussion on Population-Specific Regulation

We now include a discussion of the biological significance of population-specific gene expression patterns (Lines 526–536). The analysis shows that northern populations (e.g., Inner Mongolia and Liaoning) exhibit enhanced hormonal and metabolic regulation to cope with cold stress, while southern populations (e.g., Yunnan) rely more on antioxidant and cellular protective mechanisms. The hybrid population exhibited mixed regulatory features, suggesting high plasticity in gene regulation. These findings highlight the dual role of environmental and genetic factors in shaping diapause strategies and contribute to a better understanding of geographic adaptation in L. dispar.

1. Gray, D.R., et al., Toward a Model of Gypsy Moth Egg Phenology: Using Respiration Rates of Individual Eggs to Determine Temperature–Time Requirements of Prediapause Development. 1991(6): p. 1645-1652.

2. Bell, R.A., Manipulation of diapause in the gypsy moth, Lymantria dispar L., by application of KK-42 and precocious chilling of eggs. Journal of Insect Physiology. 1996. 42(6): p. 557-563.

3. Tauber, M.J., et al., Dormancy in Lymantria dispar (Lepidoptera: Lymantriidae): Analysis of Photoperiodic and Thermal Responses. 1990(3): p. 494-503.

4. Wei, J., et al., Impact of temperature on postdiapause and diapause of the Asian gypsy moth, Lymantria dispar asiatica. Journal of Insect Science, 2014. 14: 5.

Comment 4:

According to the authors description, transcriptomic analyses were conducted on five species. However, only one dataset (including Q30, mapping rates, etc.) is presented in the results. Why? It is advisable to include detailed data in the supplementary materials or upload them to the NCBI database for comprehensive accessibility.

Answer:

We appreciate the reviewer’s concern regarding the completeness of our data and fully understand the importance of providing comprehensive datasets to ensure transparency and reproducibility in scientific research. In the original submission, we indeed performed transcriptome sequencing for five populations of L. dispar, but only included the lowest quality metrics to briefly reflect the overall sequencing quality in the main text. This decision was originally made to maintain conciseness. However, we acknowledge that this approach may have led to insufficient detail. Therefore, in this revised version, we have made the following changes to improve data accessibility and transparency in response to the reviewer’s suggestions:

1. Addition of Supplementary Materials with Full Quality Metrics for All Five Populations

We have compiled the complete transcriptome sequencing quality statistics for all five populations into a table, which is now included as Supplementary Table 1 (File S1). This table provides details such as the number of clean reads, Q30 values, GC content, and mapped ratio for each population, offering a clear view of the sequencing quality across all samples.

2. Upload of Raw Sequencing Data to NCBI SRA with Access Information

Following the reviewer’s recommendation, we have submitted all raw transcriptome sequencing data for the five populations to the NCBI Sequence Read Archive (SRA). The corresponding SRA accession number (BioProject ID: PRJNA1238095) is now provided in the “Data Availability” section of the manuscript. The uploaded dataset includes clean FASTQ files, experimental design details, and sample metadata.

Comment 5:

In the results section, the authors described "The period from pre-diapause to mid-diapause was termed as the diapause initiation stage, whereas the period from mid-diapause to post-diapause was termed as the diapause termination stage." Which period constitutes the "diapause maintenance stage."? Additionally, the Materials and Methods section indicates that diapause egg samples were selected, yet the results lack data pertaining to the diapause stage.

Answer:

Thank you for your insightful comment. We address your concerns in two parts:

1. Clarification and Revision of Diapause Stage Definitions

We acknowledge that the original phrasing may have led to ambiguity. In L. dispar, diapause consists of three well-defined phases:

Pre-diapause: Also referred to as the preparation phase, during which embryonic development is completed in preparation for diapause.

Diapause (mid-diapause): This is the maintenance stage, characterized by developmental arrest and low metabolic activity.

Post-diapause: Also known as the termination phase, when development resumes.

We refer to the transition from pre- to mid-diapause as the diapause initiation process, and the transition from mid- to post-diapause as the diapause termination process. This terminology is now clearly explained in the manuscript (Lines 329–333):

By comparing differentially expressed genes and proteins among the three stages (pre-diapause, diapause, and post-diapause), we screened differentially expressed genes and proteins involved in the diapause initiation process (transition from pre-diapause to diapause) and the diapause termination process (transition from diapause to post-diapause).

2. Addition of Diapause Phase Data

We also appreciate your attention to data completeness. In the origin

---

## [Editor Report · Decision Letter 1]

15 May 2025

Identification of Differentially Expressed Genes and Proteins Related to Diapause in Lymantria dispar: Insights for the Mechanism of Diapause from Transcriptome and Proteome Analyses

PONE-D-24-55985R1

Dear Dr. Shi,

We’re pleased to inform you that your manuscript has been judged scientifically suitable for publication and will be formally accepted for publication once it meets all outstanding technical requirements.

Kind regards,

Munir Ahmad, PhD

Academic Editor

PLOS ONE
---

## [Editor Report · Acceptance letter]

PONE-D-24-55985R1

PLOS ONE

Dear Dr. Shi,

I'm pleased to inform you that your manuscript has been deemed suitable for publication in PLOS ONE. Congratulations! Your manuscript is now being handed over to our production team.

Kind regards,

on behalf of

Dr. Munir Ahmad

Academic Editor

PLOS ONE